# Bacterial Interactions with Nanoplastics and the Environmental Effects They Cause

Rongyu Wang [1], Xiaodong Li [1], Jing Li [2], Wei Dai [1] and Yaning Luan [1,*]

[1] The Key Laboratory for Silviculture and Conservation of Ministry of Education, College of Forestry, Beijing Forestry University, Beijing 100083, China; bjfu1999@bjfu.edu.cn (R.W.); xiaodongli@bjfu.edu.cn (X.L.); daiwei1@bjfu.edu.cn (W.D.)

[2] School of Environment and Civil Engineering, Jiangnan University, Wuxi 214122, China; jingli9411@163.com

[*] Correspondence: luanyaning@bjfu.edu.cn

**Abstract:** Recently, there has been an increase in interest in the relationship between microorganisms and micro/nanoplastics. Particularly in natural environments, bacteria play an important role. For the ecological risk assessment of plastic particles, a proper understanding of how bacteria and plastic particles interact is crucial. According to a review of the research, the interaction between bacteria and nanoplastics is primarily caused by the interaction of nanoplastics with bacterial cell membranes and the induction of oxidative stress, which can have an impact on bacterial growth, lead to alterations in biofilm production, and induce bacterial gene mutations. On a more general scale, the high concentration of nanoplastics in the environment can increase the likelihood of organic pollution reaching microbial communities, altering the gene abundance of bacteria involved in material cycling, and decreasing the activity of bacterial functional enzymes, all of which can obstruct the cycling of environmental elements. The majority of current research relies on laboratory tests, and the modeled NPs employed may be considerably dissimilar from those found in the environment. In order to provide a guide for environmental management in the future, it will be necessary to analyze the effects of nanoplastics and bacteria on the environment under actual environmental conditions to help us comprehend the relationship between nanoplastics and bacteria and their ecological impacts.

**Keywords:** viruses; nanoplastics; interaction; environmental impact





## 1. Introduction

The most widely discussed environmental issue in the world today is the use of plastics, which are polymeric materials manufactured from petroleum resources and comprise a variety of synthetic or semi-synthetic organics with high molecular weights and stable crystal structures that are challenging to degrade [1]. Plastic shards and microplastics can break down into nanoplastics (NPs) in the natural environment through the processes of hydrolysis, biodegradation, and mechanical abrasion [2]. NPs are also released into the environment through a number of biomedical applications and consumer goods that contain produced nanoparticles [3]. Despite being regarded as an incredibly durable material, plastics can break down into smaller particles. For example, they can fragment into macroplastics (>5 mm) and microplastics (MPs) (1 μm–5 mm), which can then further deteriorate into nanoplastics (NPs) (<1 μm) due to a combination of physical, chemical, and biological processes [4–6]. The physicochemical characteristics of microplastics on the verge of becoming nanoplastics can differ dramatically from those of materials with larger atom sizes. Smaller-particle-sized nanoplastics have a more specific surface area and adsorption capability, which could have a bigger influence on the environment [7]. Micro (nano) plastics can offer new habitats for microorganisms as a new ecological niche in solid environments, and their surfaces can be selectively loaded with particular bacteria. The two will inevitably interact with one another. Large concentrations of micro(nano)plastics

can have a direct or indirect impact on the composition and function of microorganism communities within the environment [8].

Microorganisms, particularly bacteria, are essential for decomposition and the regulation of biophysiological processes on Earth. When bacterial surface biomolecules are impacted by nanoparticles, environmental studies have discovered that bacteria have several opportunities to interact with nanoplastics in the environment because of their tiny size and ubiquity [9]. The study of how bacteria are affected by nanoplastics in the environment is still in its early stages. Currently, a great deal of study is required to evaluate the biosafety of nanoplastics. To facilitate further research on the influence of nanoplastics on bacteria and to elucidate the environmental behavior of nanoplastics and the mechanisms that affect bacteria, this paper reviews and summarizes relevant research on the effects of nanoplastics on bacteria from both domestic and international perspectives. It particularly focuses on the following aspects: (1) effects of nanoplastics on bacteria; (2) mechanisms through which bacteria are impacted by nanoplastics; (3) elements that affect interactions between bacteria and nanoplastics; (4) combined effects of microbes and nanoplastics on the environment. This study can act as a theoretical roadmap for research on environmental contamination and the ecological threat posed by nanoplastics.

## 2. Effects of Nanoplastics on Bacteria

### 2.1. Inhibitory Growth

Different growth inhibitory effects on bacteria are produced when nanoplastics come into contact with bacteria. For instance, co-culturing polystyrene plastics with various diameters (60, 220, 430, 700, 1040, 1700, and 2260 nm, respectively) with *Escherichia coli* (*E. coli*) results in inhibitory effects on both bacteria [10]. The findings of the experiment demonstrate that the growth inhibition of the bacteria rises with the size of the plastic fragments. The activity of both bacterial cells was most severely reduced by nanoplastics with a diameter of 1040 nm [10], effectively slowing the growth of *E. coli* and *Bacillus* because of their similarity in size to bacterial cells [10]. Both *E. coli* and the marine bacterium *Halobacterium alkalophilum* experienced growth inhibition in the presence of polystyrene nanoplastics (PS NPs) (69 nm, 5, 10, 20, and 50 mg/L) and PS NPs (50 nm, 80 mg/L), with more evident growth inhibition observed at higher concentrations [11,12]. Nanoplastics can bind to *B. hydrophila*'s cell membrane in other environments, such as wholly anaerobic systems, preventing *B. hydrophila* from growing and metabolizing [13]. In conclusion, the size and concentration of plastic fragments in various systems are directly connected to bacterial growth.

### 2.2. Biofilm Formation Impact

Nanoplastics have the capacity to differentially influence how bacteria produce biofilms. Okshevsky et al. (2020) demonstrated that NPs (20 nm) had no effect on the growth of marine bacteria but did affect bacterial biofilm formation [14]. Because the surface characteristics of NPs have a varied impact on bacteria, this has an impact on how quickly NPs aggregate [14]. Exposure to NPs also led to significant alterations in the composition of the bacterial population [14]. As a result, the community structure may change as a result of the varied impacts that nanoplastics have on the development of bacterial biofilms.

### 2.3. Genetic Linkage

Nanoplastics, smaller than microplastics but capable of adsorbing onto bacterial surfaces, can cause oxidative or mechanical damage to DNA, which can increase the number of resistant mutations in bacteria. Nanoplastics can increase the frequency of gene exchange in the environment. For instance, in aquatic ecosystems, micro/nano plastic pollution (50–1500 particles per cage) increases gene exchange between *E. coli* and *Pseudomonas* spp., with a higher frequency of plasmid transfer in bacteria that come into contact with micro/nano plastics compared to those in their natural environments or natural aggre-

gates [15]. PS NPs facilitate the conjugative transfer of antibiotic resistance genes by inducing excessive reactive oxygen species and oxidative stress, increasing cell membrane permeability, and up-regulating the expression of mating pair formation genes (*trbBp* and *traF*) [16]. Additionally, it is possible for nanoparticles to encourage bacterial drug resistance mutations. In the presence of antibiotic stress, bacterial resistance through mutations can propagate through vertical gene transfer and result in widespread resistance [17]. NPs drastically affect the number of genes associated with carbon decomposition and phosphorus cycling in bacterial communities [18]. The literature mentioned above imply that the presence of significant amounts of nanoplastics in the environment modifies the abundance of genes related to the cycling of matter and raises the likelihood of bacterial gene alterations.

### 3. Mechanisms by Which Nanoplastics Affect Bacteria

*3.1. Nanoplastics and Cell Membrane Interactions*

As listed in Table 1, in the majority of experimental trials, the negative effects caused by bacteria were linked to the interaction between nanoplastics and bacterial cell membranes. Nanoplastics interact with cell membranes in a manner that is comparable to that of ordinary nanoparticles (such as metal oxide nanoparticles) [19].The way that bacteria react to nanoplastics depends on the composition of their cell membranes [13]. As the amount of attachment grows, micro/nanoplastics can cause cell wall indentation in microbial cells, which may result in cellular dysfunction [20]. For instance, nanoplastics directly interacted with the cell membrane and extracellular polysaccharides (EPs) of the anaerobic bacterium *Shewanella oneidensis* in both aerobic and anaerobic conditions, changing the bacterial riboflavin secretion [21]. Crushed-grained molecular simulations revealed that lipid bilayers' enhanced nanoplastic permeability was the cause of changes in cellular activities [22].

Nanoplastics and bacterial membranes interact differently depending on their sizes. For instance, it was discovered that different sizes of polystyrene plastic particles influence their interactions with the membrane of the Gram-negative bacterium *E. coli*, with 0.02 μm PS showing the strongest adsorption to the cell surface, followed by 0.2 μm PS, whereas no adsorption of 2 μm PS on the cell surface was observed because interactions between larger plastic particles and bacterial cells are more repulsive [23]. Smaller nanoplastics, particularly those smaller than 100 nm, can interact with the surface receptors of bacteria, changing the shape of the membrane and causing bacteria to internalize nanoplastics through endocytosis [24].

Additionally, NPs with various charges have a more profound impact on interactions between bacterial cell membranes. The Gram-negative halophilic proteobacterium *Halomonas alkaliphila* experienced membrane degradation as a result of unmodified and amine-functionalized nanoplastics (50 μm) [11]. Perini et al. (2022) noted that amine-functionalized NPs had greater membrane disruption in lipid bilayers than carboxyl-functionalized NPs [25]. Dai et al., investigated the action characteristics and transmembrane endocytosis mechanism of nanoplastics with different charges on bacterial cell membranes, and the results showed that the entry of nanoplastics into bacteria is dependent on the surface charge of the nanoplastics and the structural characteristics of cell membranes. Positively charged polystyrene nanoplastics can be efficiently translocated across the cell membrane, whereas negatively charged and neutral nanoplastics face difficulties in efficiently crossing the membrane and entering cells. Consequently, amine group-functionalized NPs are more effective against *E. coli*, and the toxicity of *Bacillus* sp. is higher than that of carboxyl-functionalized NPs and unmodified NPs [26]. However, no information has been provided regarding what happens to nanoplastics after they penetrate a cell membrane. In conclusion, nanoplastics can directly contact cell membranes, and their interaction with bacterial cell membranes becomes more prominent as the size of the amine-functionalized nanoplastics decreases.

**Table 1.** Summary of the effects of nanoplastics on bacterial production.

| Types | Size | Concentrations | Time | Mechanisms of Interaction | Ref. |
|---|---|---|---|---|---|
| *Staphylococcus aureus*, *Bacillus subtilis*, *Escherichia coli* | PCLA: 50 nm HDPE: 615 nm PESa: 396 nm PP: 531nm | 10–100 mg/L | 24 h | Decreased lipid peroxidase activity and increased antioxidant ratio lead to the occurrence of oxidative stress | [27] |
| *Pseudomonas aeruginosa* PAO1 | PS: 120 nm | 0.1, 1, 10, 20, 50 mg/L | 72 h | ROS generation, oxidative stress induction, membrane interaction | [28] |
| *Methanosarcina acetivorans* | PS-NH$_2$: 30 nm | 5, 8 mg/L | 4 days | ROS generation, oxidative stress induction, membrane interaction | [29] |
| *Methanosarcina acetivorans* | PS-NH$_2$: 30 nm | 20, 50 mg/L | 4 days | ROS generation, oxidative stress induction, membrane interaction | [29] |
| *Alkalophilic halomonas* | PS: 50 nm PS-NH$_2$: 55 nm | 80 mg/L | 0.5–2 h | ROS generation, oxidative stress induction, membrane interaction | [11] |
| *Lactobacillus plantarum* ZP-6 *Lactobacillus brevis* ZB-1 *Lactobacillus fermentum* ZF-3 *Lactobacillus gasseri* ZG-4 | PP: 100 nm PE: 100 nm PVC: 100 nm | 30 mg/L | 2 h, 4 h, 6 h, 8 h | Membrane interactions were observed | [30] |
| *Sylvatica* | PS: 159nm | 150 mg/L | 4 h–5 day | Membrane interactions were observed | [21] |
| *Escherichia coli* | PS: 200nm | 4 mg/L | 24 h | Membrane interactions were observed | [23] |

*3.2. Nanoplastics Induce Oxidative Stress in Bacteria*

　　Cell oxidative stress results from the interaction of nanoplastics with cell membranes. One of the most common ways that nanoplastic toxicity manifests in organisms is through the development of oxidative stress [31]. In particular, nanoplastics enter the body through endocytosis and use NADPH oxidase to produce reactive oxygen species (ROS). More ROS may be produced as a result of stronger interactions; however, excessive ROS levels considerably reduce bacterial viability.

　　Bacteria are stimulated by oxidative stress to create more EPS, enzymes, and other components of biological reactions [11]. For instance, when *H. alkaliphila* was exposed to 80 g/mL of unmodified or amino-modified nanoplastics (50 nm), there was a significant increase in EPS secretion and high oxidative stress, which decreased survival of bacteria [11]. By increasing ROS generation, nanoplastics in high concentrations (80 mg/L) reduce the efficiency of inorganic nitrogen conversion in the marine bacterium *H. alkaliphila* [11]. At a relatively high concentration (200 g/mL), Chen et al., (2020) found that carboxyl-modified nanoplastics increased bacterial ROS production while decreasing hydrolase activity and cell surface charge [32].

　　Additionally, the antioxidant activity against traditional plastics and bioplastics was altered in the Gram-negative bacteria Pseudomonas aeruginosa and *E. coli* [27]. These findings imply that the interaction of bacterial membranes with nanoplastics leads to a series of negative effects, most of which are mediated by oxidative stress. Although this mostly happens in environments with high amounts of nanoplastics, isolating nanoparticles with EPS can help bacteria defend themselves [11]. In general, interactions between bacterial cells and nanoplastics result in membrane rupture, which lowers hydrolase activity and surface charge, produces more EPS, and increases cellular oxidative stress. Figure 1 shows a pictorial representation of the mechanisms by which nanoplastics impact bacteria.

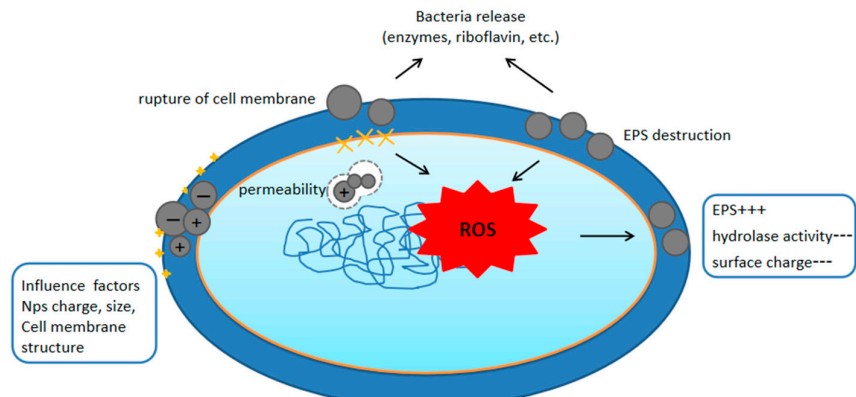

**Figure 1.** Mechanisms by which nanoplastics affect bacteria. "+++"implies an increase, "---"Implies a reduction.

## 4. Factors Affecting the Interaction of Nanoplastics with Bacteria

### 4.1. Properties of Plastic Pellets

The polymer composition, size, shape, color, and functional groups of MPs/NPs in natural settings vary. These characteristics of MPs/NPs are crucial for assessing how they interact with bacteria since they impact the plastic's adsorption capacity, bioavailability, and cellular uptake.

Smaller NPs have significant physicochemical characteristics that set them apart from MPs, including Brownian motion, a greater specific surface area, and a stronger interaction with other pollutants [33]. The surface activity, surface energy, surface charge, and electronic structure of nanoparticles (NPs) are influenced by their size, which also changes the types of forces that interact with nearby biological surfaces [34]. For instance, smaller NPs are more likely to combine under specific conditions compared to bigger particles due to increased surface energy [35]. The smallest NPs (50 nm) encourage aggregation to a greater extent than bigger NPs (100 and 500 nm), according to Summers et al., (2018), which can result in complex interaction outcomes [36]. Song et al., also demonstrated that, in the same aquatic environment, 50 nm NPs could effectively aggregate with organic matter and metal ions while 200 nm MPs were more stable and did not form aggregates [37]. In living things, the particle size-dependent effects of NPs have the potential to cause biological neurotoxicity. Developmental and neurotoxic effects were seen in PS-NPs of all particle sizes, with smaller PS-NPs demonstrating more toxicity [38]. Furthermore, due to their increased specific surface area, enabling more interactions with cellular components, smaller nanoplastic particles are more likely to infiltrate cells and cause more severe cellular damage [23].

By generating electrostatic, spatial site resistance, and hydrophilic repulsion between polymer particles on the surface of nanoplastics, stability can be improved. The zeta potential of NPs may decrease as biomolecule concentration increases when NPs and biomolecule have opposing net surface charges, which may result in aggregate formation [39].

In complex settings, plastic particles are prone to aging, which can result in physical and chemical changes to their surfaces. Aging increases the surface area and roughness of the particles, and damages the molecular connections between the polymers in tiny plastics [40]. The ability of plastic particles to adsorb molecules is affected by their age, and it has been demonstrated that photoaged PS-MPs have lower adsorption coefficients for hydrophobic molecules, such as nonpolar aliphatic, unipolar aliphatic, bipolar aliphatic, nonpolar aromatic, and bipolar aromatic molecules [6]. While the growth and adhesion of biofilms to plastics in oceanic environments decline, the production of polymer components in the extracellular matrix of marine bacteria increases, enhancing cellular attachment to rough surfaces [41]. The interaction between NPs and EPS made by bacteria becomes more intense as they age. After aging in EPS-containing media for 12, 24, and 48 h, negatively

charged nanoplastics, as opposed to positively charged ones, show a greater binding affinity for EPS components [42].

In addition to size, aging, and surface features, NPs' shape, aggregation, and dispersion behavior, as well as surface characteristics such as polymer type, composition, surface charge, and functional groups, have an impact on how well they interact with biomolecules [43]. Numerous studies have confirmed that NPs have smaller sizes, opposing charges, and generate aging, which are critical factors impacting the circumstances for bacterial contact.

### 4.2. Properties of the Medium

The characteristics of the medium, particularly aquatic settings, can significantly affect the properties of NPs in natural environments [44]. For instance, the medium's pH, salinity, and ionic strength not only influence how NPs interact with biomolecules but also have a significant impact on how NPs behave in aquatic environments, such as their effective charge, aggregation state, hydrodynamic dimensions, surface valence, and dispersion [45].

The way that nanoplastics and microorganisms interact is influenced by the medium's salinity. The plastic's surface charge will be neutralized if the salinity in the environmental medium is too high, which will compress the plastic's electric double layer and lessen the electrostatic interactions the plastic produces during the adsorption process. Additionally, high salt concentration causes the salting-out effect, indirectly affecting the interactions [46]. Salinity also influences the physicochemical characteristics and aggregation of nanoparticles, and it has been demonstrated that high salinity can result in significant aggregation and deposition of NPs in sediments, raising the potential risk to a variety of organisms in the sediments [47]. The Derjaguin-Landau-Verwey-Overbeek interaction energy spectrum shows that low salinity increases NPs-NPs repulsion and promotes the establishment of a stable state. A drop in repulsive forces and a reduction in the net repulsive energy barrier between NPs are caused by the electrostatic double layer compressing as salinity rises. Salinity significantly affects the stability, mobility, toxicity, as well as the ecological effects of NPs in various environmental contexts, resulting in NP aggregation.

By altering their surface charge, electrolytes in an aqueous environment can have an impact on how NPs aggregate [48]. For instance, NPs displayed a stable state at 0.01 mM $FeCl_3$, but they exhibited aggregation at $FeCl_3$ concentrations of 0.1 and 1 Mm [49]. The valence state of the ions is a crucial element governing how plastic particles aggregate in aquatic environments. For instance, the critical coagulation concentration (CCC) of PSNPs in a monovalent NaCl solution is 32 and 80 times greater than in a divalent $MgCl_2$ and $CaCl_2$ solution [50], respectively.

The ability of NPs and other tiny polymers to adsorb additional pollutants is also substantially influenced by the pH of the media. The ability of sulfamethoxazole to bind to polyethylene, polystyrene, polyethylene terephthalate, polyvinyl chloride, and polypropylene was found to decrease with increasing pH [46]. This finding suggests that pH may also influence how plastic particles interact with microorganisms. The aggregation behavior of NPs is ultimately controlled by electrostatic forces through the surface charge of the plastic particles and the ionization of their functional groups, both of which are influenced by pH [51]. With the pH increases in a NaCl solution, both untreated nanoplastics (100 nm) and carboxyl-modified nanoplastics (303 nm) become more stable [52]. To summarize these results, the interaction between NPs and bacteria may be stronger at higher salinity, electrolyte concentration, with lower ionic valence, but the pH of the medium does not appear to have a discernible pattern.

## 5. Ecological Impacts of Nanoplastics and Bacteria Working Together

### 5.1. Nanoplastics Affect the Environment by Influencing Bacteria

Micro/nanoplastics may interact with creatures through ingested consumers due to their low biodegradibility, which can encourage the buildup of persistent organic pollutants (POPs) in the food chain [53]. Due to their larger surface area, micro/nanoplastics

have a higher capacity to absorb and desorb hazardous chemicals [54]. They can also impact bacterial biofilm growth and, consequently, the underlying microorganisms. For instance, polystyrene microplastics influence the assemblage of epipelagic microbes as well as biofilm growth [55]. Nanoplastics may hinder the movement of other pollutants through soil or water and even cause antagonistic behavior, which increases their toxicity to nearby organisms.

Nanoplastics alter bacterial populations in the digestive tract. For instance, when nanoplastics were introduced into a pure anaerobic digestion system, the growth and metabolism of Acinetobacter hydrophilus were partially suppressed. In mixed anaerobic digestion systems, nanoplastics severely hindered the anaerobic digestion of sewage sludge and increased the start-up time of the mixed anaerobic digestion system [13]. Additionally, methane production performance, such as cumulative methane production, maximum daily methane production, and lag phase time, was hindered by the presence of nanoplastics in the system [13]. Reduced metabolic toxicity was seen in thermophilic bacteria when amine-functionalized polystyrene nanoplastics and PFOS were mixed, which led to enhanced oxidative stress, increased cell permeability, and decreased hydrogen production [32]. Additionally, nanoplastics impact bacterial biotransformation processes. The marine bacterium *Bacillus alkaliphilus*' chemical makeup and ammonia conversion efficiency was impacted by exposure to NPs [11]. PS NPs may also negatively impact the capacity of anaerobic bacteria to convert biomass into energy, which could impact biodiversity and biogeochemical cycling in both natural and manmade ecosystems [18].

The elemental cycle is disrupted in varying degrees by the widespread use of micro/nanoplastics. These micro/nanoplastics, a specific kind of organic carbon, can be found in a range of environmental media. The carbon cycle is driven by microbial communities, and the presence of micro/nanoplastics can change the variety, structure, and abundance of these microorganisms. For instance, microplastics cause soils to emit more $CO_2$ as a result of their interactions with active microbial communities, which, in turn, increases hydrolytic enzymes activities [56]. In another study, PS NPs were found to have different effects on soil biomass and microbial activity [57]. Low-density polyethylene microplastics, as shown by Huang et al., (2019), have an impact on the composition of microbial communities and urease activity in soil [58]. Additionally, some academics have discovered that PE microplastics change the proportion of gram-positive to gram-negative bacteria [59]. By preventing the functioning of enzymes involved in carbon conversion, micro/nanoplastics can impact the carbon cycle.

Several recent studies describe that the wide distribution of nanoplastics in the environment increases the probability of organic pollution entering the microbial community, thus greatly increasing the uncertainty of environmental pollution, but also directly affecting bacteria, thus weakening the material energy conversion within the environmental digestive system. More importantly, it is important to note that nanoplastics can affect the proportion of bacteria, bacterial functional enzyme activity, and interfere with the environmental cycle of the elements.

*5.2. Nanoplastics and Bacteria form Ecological Corona That Affects the Environment*

The interaction of nanoplastics with macromolecules in extracellular polymeric substances is unavoidable, since extracellular polymeric substances, which are mostly made by microbes, are the primary source of natural organic matter in the environment. EPS (the product of cell lysis and hydrolysis, mostly comprised of proteins, polysaccharides, humic substances, DNA, and other things) makes up the majority of the combination of macromolecules released by microorganisms, such as bacteria, throughout their growth process [60]. Consequently, the EPS produced by bacteria is the first to interact with nanoplastics.

Several studies have explored the formation of ecological corona by nanoplastics and EPS in the environment and its effects (Table 2). Through electrostatic and hydrophobic interactions, NPs bind and adsorb endogenous proteins to produce a biological corona [43]. In contast, the ecological corona, is an abiotic phenomenon that shows interactions between

NPs and a particular set of EPSs that may be modified by nearby environmental elements and result in exposure scenario-related effects. The bioavailability of NPs can be greatly changed by the development of the ecological corona, which is an important factor affecting their essential biological effects [43]. The ecological corona takes in macromolecules from its environment and controls how NPs interact with and exchange with minerals, organic matter, and contaminants in the aquatic body. The biological and chemical reactivity of NPs may thus be determined by the ecological corona that has been acquired [42].

The mobility of nanoplastics in saturated porous media is also determined by the eco-corona. The surface functions and macromolecular source of nanoplastics have a substantial impact on the characteristics of eco-corona and its capacity to improve nanoplastic transport [60]. More so than eco-corona derived from humic and xanthic acids in soil, ecocorona derived from gram-negative *E. coli* EPS improves the transport of polystyrene nanospheres in saturated porous media [60].

**Table 2.** Formation of ecological corona by nanoplastics with EPS in the environment and the results of its influence.

| Types | Size | Concentrations | Time | Ecocorona Composition | Aims | Ref. |
|---|---|---|---|---|---|---|
| *Halomonas genus* | 50–500 nm | 5 mg/L | 24 h | Glycoprotein polymers | The generation of NPs-EPS ecological corona affects the sinking rate. The EPS composition may affect the buoyant density of NPs' agglomerates. | [36] |
| *Chlorella vulgaris* | 200 nm | 1 mg/L | 72 h | 20 kDa to >100 kDa in size | EPS further enhanced the positive effect of PS MPs on bacterial growth. | [42] |
| *Marine diatoms* | 60 nm | 1–50 mg/L | 3 h | Carbohydrates and proteins | EPS increased the colloidal stability of PS M/NPs in NaCl or CaCl$_2$. | [44] |
| *Escherichia coli* | 20–50 nm | 200 mg/L | 40 min | Proteins dominate over polysaccharides in the corona | EPS corona formation readily adsorbs metal ions, forming NPs-EPS-metal complexes and promoting sedimentation. | [61] |
| *Escherichia coli* | 500nm | 10 mg/L | 40 min | Approximately 70% of the adsorbed HA macromolecules were over 30 kDa | EPS corona enhances the transport of nanoplastics. | [62] |
| *Bacillus subtilis* | <1000 μm | 10 mg/L | 40 min | Glycoprotein polymers | The NPs-EPS corona increased the toxicity of NPs to the environment by accumulating highly toxic heavy metals. | [63] |

## 6. Conclusions

The smaller-scale effects of nanoplastics on bacterial development in the environment include decreased growth activity, decreased enzyme activity, decreased cell surface charge, and changed biofilm formation patterns. The primary cause of this occurrence is the possibility of nanoplastics attaching to microbial cell membranes, resulting in cell wall depression, initiating oxidative stress and potentially resulting in cellular malfunction.

On a large scale, the presence of large amounts of nanoplastics in the environment increases the likelihood that organic pollution will enter the microbial community, also increasing the likelihood that bacterial genes will be mutated, affecting the proportion of bacteria, changing the abundance of genes related to the cycling of matter, and decreasing the activity of bacterial functional enzymes, thereby interfering with the environmental cycle of elements. Additionally, nanoplastics directly impact microbes, weakening the environment's digestive system and impairing the conversion of materials into energy,

among other effects. Most current research is focused on lab tests with model NPs that may be substantially dissimilar to those found in the environment.

Therefore, the following areas need to be strengthened in future studies:

(1) In order to mimic and illustrate the toxicity of nanoplastics in true natural habitats, it is advised that future research concentrate more on the toxicity of nanoplastics at realistic concentrations in natural environments;

(2) Nanoplastics are materials with extremely small particles that are chemically similar to organic matter. Future research should be done to fill the gap in this field. There is only one experimental way to study the interactions between nanoplastics and bacteria and it is difficult to directly detect the distribution and abundance of nanoplastics in the environment due to shortcomings in detection methods and tools;

(3) To gather accurate information on the toxicity of NPs to microorganisms, it is important to examine the kinetics and absorption pathways of various MP types (polymer type, size, and shape) when interacting with various bacterial species.

**Author Contributions:** Conceptualization, R.W. and Y.L.; methodology, Y.L. and J.L.; investigation, R.W.; writing—original draft preparation, R.W.; writing—review and editing, Y.L., X.L., J.L. and W.D.; visualization, R.W. and X.L.; supervision, Y.L.; project administration, Y.L.; funding acquisition, Y.L. All authors have read and agreed to the published version of the manuscript.

**Funding:** This study was supported by the National Natural Science Foundation of China (Grant No. 32001197).

**Institutional Review Board Statement:** Not applicable.

**Informed Consent Statement:** Not applicable.

**Data Availability Statement:** All data have been included in the main text.

**Acknowledgments:** We thank all members for their discussion during the preparation of this manuscript.

**Conflicts of Interest:** The authors declare no conflict of interest.

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
