# Peer review of "Bacterial Interactions with Nanoplastics and the Environmental Effects They Cause"

_fermentation, doi:10.3390/fermentation9110939_

Round 1

Reviewer 1 Report

This review discussed the effect of nanoplastics on the environment and the related bacterial interactions. The paper is well organized and current trends are well presented. However, several issues need to be addressed before acceptance.

1.       Authors are suggested to give the definition of nanoplastics, and potential readers would get cleaner information of this paper. Currently, the size range of nanoplastics is different in various reports (Water Research, 2022, 221, 118846.; Science of the Total Environment 2022, 846, 157371.; Water Research, 2020, 183, 116046.), and authors are suggested to discuss this issue.

2.       Line 82, “Nanoplastics, which are larger than microplastics”, this sentence is not correct.

3.       The effect of size on the properties of nanoplastics should be further enhanced, several references should be included in the paper, Water Research X, 2023, 19, 100169.; Science of The Total Environment, 2023, 872, 162096.

4.       Authors need to improve their language, and concentration units are not the same (mg/l or mg/L?) and there are too many sentences start with “According to…”. Also, the citation format does not follow the requirement of the journal.

5.       What is EPS, provide the full name. Check this issue in the whole manuscript.

6.       Authors should update the references, and papers from the last three years are suggested to make the presented information more relevant and accurate.

Authors need to improve their language, and concentration units are not the same (mg/l or mg/L?) and there are too many sentences start with “According to…”. Also, the citation format does not follow the requirement of the journal.

Reviewer 2 Report

The work submitted to evaluation more than a research work is a work of compilation of studies carried out by different authors. It does not represent an advance in the knowledge of the field in which it is framed, but it is a great work tool for those who start in the subject of interactions and processes of degradation, adsorption, etc. of microplastics.

It is also a great tool for teaching and teaching courses at undergraduate and graduate level.

As an inconvenience to indicate that the authors do not contribute anything new for the most versed researchers in this field.

Reviewer 3 Report

The review paper manuscript “Bacterial Interactions with Nanoplastics and the Environmental

Effects They Cause

Overall, this is a well written manuscript and has a potential to be accepted.

Nevertheless, the authors should revise better their Conclusion. The study must summarize and clearly present the findings related to previous research in this area.

In addition, minor  comments follow.

1) Lines 49-54: Is proposed to be rephrased.

2)Lines 50-51: It is not clear the difference between (1)”…………polymeric nano plastic” and (2) “…..nano plastic”. Do you mean microplastic?

3) Lines 55-71: Please clarify/state whether an increase in concentration /size always connected to growth inhibition.

4) Lines82-83: It seems that a typo error occurs. “Nanoplastic…..larger than microplastic…..”

5) Please improve Table 1 as well as Table 2..

6) Please indicate the origin of figure 1 and include  reference of figure 1 it in the text in order to be clear the purpose of its presence .

7)Please clarify the term “natural ambient concentration”

8) Please extend and update references (including 2023 findings where applicable)

I will be glad to provide further details if needed and thank you for contacting me.

Minor editing of English language required

Round 2

Reviewer 1 Report

accept!

Author Response

We appreciate your careful attention to our manuscript. Thanks very much for your help and time.